

# Mesoscale modulation of marine boundary layer water vapor isotopologues during EUREC4A

Joseph Galewsky[1] and Sebastian A. Los[1]

[1]Department of Earth and Planetary Sciences, University of New Mexico, Albuquerque, New Mexico
**Correspondence:** Joseph Galewsky (galewsky@unm.edu)

**Abstract.** Shallow cumulus clouds in trade-wind regions remain a major source of uncertainty in climate projections, with conflicting hypotheses about how mesoscale circulations affect boundary layer moisture. We analyze water vapor isotopologue measurements from the EUREC[4]A campaign to quantify mesoscale modulation of marine boundary layer humidity and composition. Surface $\delta$D measurements from R/V Meteor show remarkably strong sensitivity to mesoscale vertical motions, responding 7.5 times more strongly than humidity when normalized by observed standard deviations. Mesoscale upward motion counteracts entrainment-driven isotopic depletion with an efficiency of 1.19, meaning 1 mm s$^{-1}$ of vertical velocity more than cancels the isotopic effect of 1 mm s$^{-1}$ of entrainment. The strongest correlations between vertical velocity and both $\delta$D ($r \approx 0.52$) and mixing ratio ($r \approx 0.39$) occur within $\pm 200$ m of the subcloud layer (SCL) top. A flux-form mixed-layer model reproduces these asymmetric responses, providing mechanistic understanding of how mesoscale circulations fundamentally modulate boundary layer moisture processes.

## 1 Introduction

Shallow cumulus clouds in the trade-wind regions are ubiquitous and exert a cooling influence on the climate, but their response to warming remains highly uncertain (Bony and Dufresne, 2005). These low clouds have long been recognized as a leading source of spread in climate model projections of global warming (Sherwood et al., 2014). Many climate models predict a positive trade cumulus cloud feedback governed by reductions in cloud fraction near cloud base. In particular, higher-sensitivity models tend to produce more efficient entrainment of dry air from aloft, which depletes low-level humidity and erodes cloud cover (Sherwood et al., 2014). This hypothesized mixing–desiccation mechanism posits that vigorous shallow convective mixing dries the lower troposphere and dissipates clouds, thereby amplifying surface warming as a positive low-cloud feedback.

Recent observations, however, challenge this simple picture. In early 2020, the EUREC[4]A field campaign (Elucidating the Role of Clouds–Circulation Coupling in Climate) was conducted near Barbados with a comprehensive network of research aircraft, ships, and ground stations to study trade-wind cumulus and their environment (Bony et al., 2017; Stevens et al., 2021). Analyses of EUREC[4]A data revealed ubiquitous shallow mesoscale circulations on scales of roughly 100–200 km that organize convection and concentrate moisture in the trades (George et al., 2023). Consistent with this, periods of stronger mesoscale ascent did not lead to a drier subcloud layer (SCL) or reduced cloudiness, contrary to the mixing–desiccation expectation (Vogel et al., 2022). Instead, the observations suggest that mesoscale cloud–circulation coupling can maintain humidity, implying





that factors beyond one-dimensional entrainment, such as horizontal convergence and large-scale vertical motion, significantly influence low-level moisture and cloud cover. Nevertheless, disentangling the contributions of these processes, for example separating the effects of shallow convective detrainment from those of large-scale subsidence, remains challenging with conventional measurements alone. Standard thermodynamic observations cannot easily attribute moisture variability to specific

physical processes, leaving an important gap in process-level understanding of the trade cumulus regime.

Stable water isotopologues offer a way to fill this gap. The ratios of heavy to light water isotopologues in vapor, such as $H_2^{18}O/H_2^{16}O$ or $HDO/H_2O$, commonly reported as $\delta^{18}O$ and $\delta D$, are sensitive to the cumulative phase-change history of an air mass (Galewsky et al., 2016). Condensation and rainout preferentially remove heavy isotopes, so air that has undergone extensive convective uplift and precipitation is left isotopically depleted in heavy molecules relative to ocean water. In contrast,

addition of moisture by evaporation from the warm ocean surface, or by mixing with unsaturated air from below, enriches the vapor in heavy isotopes. Thus, water vapor isotopic measurements can serve as tracers of moisture origin and transport. They enable us to distinguish between air masses that have experienced different water-cycle processes and to test hypotheses about what controls an air mass's humidity and cloud-forming potential. Past studies have demonstrated that isotopic variations can be used to identify moisture sources and quantify mixing between atmospheric layers, processes that are largely indistinguishable

in bulk humidity alone (Risi et al., 2019; Galewsky et al., 2016).

Motivated by this, EUREC$^4$A included a coordinated initiative (EUREC$^4$A-iso) to deploy an extensive network of water vapor isotope analyzers across multiple platforms (Bailey et al., 2023). Seven laser-based instruments, sampling at up to 0.5 Hz, were operated on two research aircraft, three ships, and at the Barbados Cloud Observatory during the campaign. This data set provides the high-resolution, multi-platform coverage needed to close regional moisture budgets and rigorously

evaluate model simulations of moist processes. Here, we leverage the EUREC$^4$A water vapor isotopic measurements from the R/V Meteor to bridge the gap between documented mesoscale circulation effects and their impacts on the marine boundary layer moisture budget. Specifically, we use the stable isotope signatures to directly track shallow convective mixing and its influence on lower-tropospheric moisture in the trade-wind regime.

## 2 Data

EUREC4A comprised roughly 5 weeks of measurements in the downstream winter trades of the North Atlantic, eastward and southeastward of Barbados. The campaign deployed an extensive observational network to characterize processes operating across scales from micrometers to hundreds of kilometers. The measurements included 2500 atmospheric soundings to quantify mesoscale and larger-scale circulations, approximately 400 flight hours by four research aircraft, operations from four research vessels, and continuous observations from a ground-based cloud observatory. Additional platforms included autonomous sys-

tems that collected nearly 10,000 upper-ocean profiles, continuous atmospheric boundary layer measurements, air-sea interface observations, water vapor isotopologue measurements across multiple platforms, coordinated satellite observations, and support from high-resolution numerical weather and climate models.





## 2.1 Upper Air Data

The High Altitude and Long Range Research Aircraft (HALO) dropsonde measurements were conducted as a core part of
the EUREC4A field campaign. The primary scientific motivation for these measurements was to characterize the mesoscale
meteorological environment of trade-wind cumulus cloud fields and to quantify area-averaged vertical motion in the marine
boundary layer. The cornerstone of the HALO measurement strategy was the EUREC4A-circle (Figure 1), a standardized
circular flight pattern designed to enable accurate estimates of mesoscale circulation properties including divergence, vorticity,
and vertical velocity. This approach, adapted from Lenschow et al. (1994, 2007) and Bony and Stevens (2019), exploits the
assumption that atmospheric variations at the mesoscale can be approximated as linear in horizontal space over the scale of a
single aircraft circle.

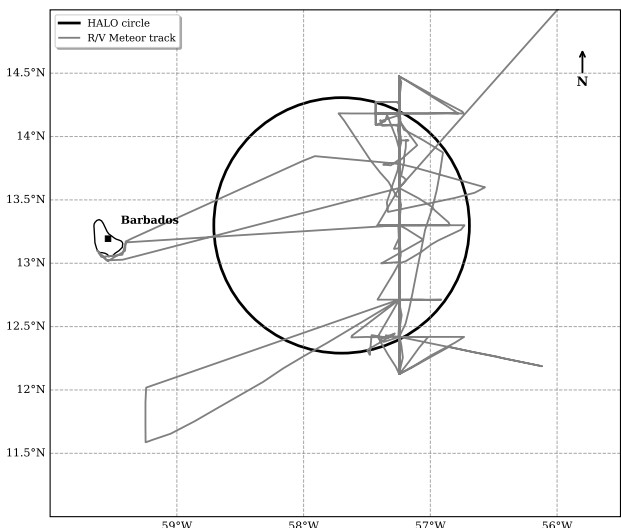

**Figure 1.** Map of the EUREC4A study area showing a representative HALO circle pattern and the R/V Meteor ship track. The circle (black)
represents one of the 222 km diameter flight patterns used for dropsonde deployment to measure mesoscale circulation. The ship track (gray)
shows the path of R/V Meteor during the January–February 2020 field campaign.

The EUREC4A-circle was centered at 13.30°N, 57.72°W with a diameter of approximately 222 km. Each circle flight pattern
deployed 12 dropsondes at regular intervals around the circumference, with launches separated by approximately 5 minutes
corresponding to the aircraft's 60-minute circuit time. This systematic sampling enables estimation of horizontal gradients
through regression analysis, from which mesoscale circulation properties can be derived. A total of 70 EUREC4A-circles were
flown by HALO during the campaign, providing unbiased sampling across meteorological conditions. HALO flights operated
from Bridgetown, Barbados, at altitudes between 10.0–10.5 km. A total of 895 Vaisala RD-41 dropsondes were launched,
each containing pressure, temperature, and humidity sensors sampling at 2 Hz, along with GPS receivers providing wind
measurements at 4 Hz. The measurements are archived in the JOANNE dataset (George et al., 2021), from which we use



the Level 4 circle-averaged products containing mesoscale diagnostics derived through regression analysis of the 12-sonde circle patterns. These products include horizontal gradients of atmospheric variables and derived quantities such as divergence, vorticity, and vertical velocity, with associated uncertainty estimates from the regression fitting.

The subcloud layer height $h$ used here was first presented in Vogel et al. (2022) and is determined using a temperature-based threshold method (Albright et al., 2022). Specifically, $h$ is defined as the height where the virtual potential temperature $\theta_v$ first

exceeds its density-weighted mean value (computed from 100 m up to $h$) by a fixed threshold $\epsilon = 0.2$ K (Vogel et al., 2022). This approach follows established methodology for identifying the top of the well-mixed subcloud layer (Touzé-Peiffer et al., 2022). The method accounts for the finite thickness of the transition layer separating the mixed layer from the cloud layer above (Albright et al., 2022). This transition layer, approximately 150 m thick, complicates the application of classical mixed-layer theory which assumes an infinitesimally thin inversion.

The entrainment rate $E$ is computed using a modified flux-jump model (Albright et al., 2022; Vogel et al., 2022) that extends the classical approach of Lilly (1968) and Stull (1976). The entrainment rate represents the deepening of $h$ due to small-scale mixing at the subcloud layer top. This approach differs from the zero-order jump models that assume instantaneous transitions (Lilly, 1968) by accounting for the finite depth of the entrainment zone, providing a more realistic representation of the actual atmospheric structure observed during EUREC4A. Note that throughout this study, $E$ refers only to entrainment from the

cloudy layer (CL) into the subcloud layer (SCL). It does not include exchange between the CL and the free troposphere (FT).

### 2.2 Isotopic Data

Water vapor isotope measurements aboard the R/V Meteor were obtained using a Picarro L2130-i cavity ring-down spectrometer (CRDS) operating at 1 Hz resolution from January 18 to February 22, 2020. The analyzer was housed in the Air-Chemistry Laboratory at $\sim$20.3 m above sea level, sampling ambient air through a 5 m long, 4.6 mm ID PTFE inlet line heated to 45°C

and insulated with polyethylene foam. The inlet was housed in a downward-facing funnel to minimize contamination from rainwater and sea spray, and included a 0.2 $\mu$m PTFE aerosol filter. Flow was controlled by the CRDS system at approximately 0.03 slpm, resulting in a time delay of >2 minutes from intake to analyzer. Daily calibration checks were performed using four liquid water standards spanning $\delta^{18}$O values from $-20.97$ to $-2.79$‰ and $\delta$D values from $-158.13$ to $-13.12$‰, delivered in gas phase using a Picarro Standards Delivery Module. Prior to normalization, isotopic observations were corrected for small

humidity-dependent biases of up to 0.24‰ in $\delta^{18}$O and 0.36‰ in $\delta$D. Total uncertainties were estimated at 0.29‰ for $\delta^{18}$O and 1.24‰ for $\delta$D by summing in quadrature the uncertainties associated with liquid standards, humidity-dependence correction, calibration measurement precision, and temporal drift over the campaign. Further details on the isotopic data collection program can be found in Bailey et al. (2023).

### 3 Results

Figure 2 presents the full time series of surface water vapor measurements from the R/V *Meteor* during EUREC4A. The dataset exhibits remarkably low variability across all measured parameters, reflecting the stable trade wind conditions that character-



ized the campaign period. Water vapor mixing ratios varied with a standard deviation of only 0.832 g kg$^{-1}$ around a mean of 15.1 g kg$^{-1}$, while $\delta$D showed a standard deviation of 1.94‰ around a mean of $-70.6$‰. Surface meteorological conditions were similarly constrained, with relative humidity varying by 4.6% around a mean of 71.6% and sea surface temperatures

showing minimal variation (standard deviation 0.23°C) around 27.3°C. The measurement uncertainty for 6-hour averaged isotope data (1.24‰ for $\delta$D) represents approximately 65% of the observed natural variability, constraining our ability to detect weak atmospheric signals. Consequently, our analysis focuses on the strongest and most robust correlations that exceed this measurement noise threshold.

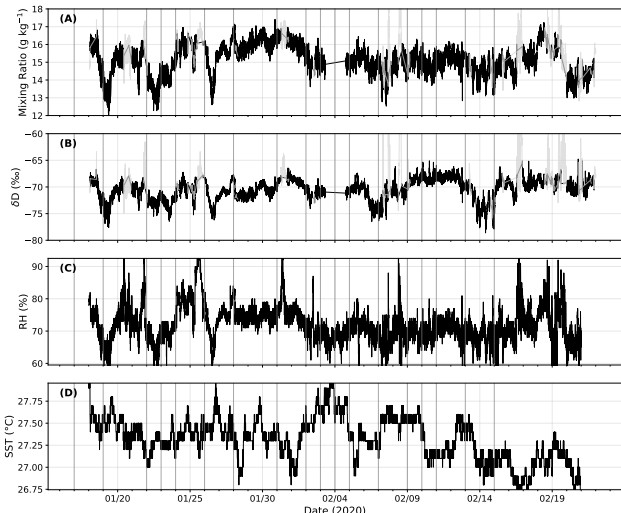

**Figure 2.** Time series of surface measurements from R/V *Meteor* during EUREC4A, January 18–February 22, 2020. (A) Water vapor mixing ratio from Picarro L2130-i cavity ring-down spectrometer at 1-minute resolution. (B) Deuterium isotope ratio ($\delta$D) from the same instrument. (C) Relative humidity from shipboard meteorological sensors. (D) Sea surface temperature. Black lines indicate quality-controlled data; pale gray points show measurements flagged during quality control. Gray vertical lines mark days with HALO aircraft circle flights from the JOANNE dropsonde dataset.

Figure 3 shows the vertical correlation structure between HALO-derived vertical velocity and surface isotopic composition

($\delta D$) and mixing ratio as a function of altitude relative to the top of the subcloud layer. Both $\delta$D (solid line) and mixing ratio (dashed line) exhibit their strongest correlations just below this boundary. The isotopic signal shows peak correlation ($r \approx 0.52$) about 100 meters below the top of the SCL, while humidity correlations reach maximum strength ($r \approx 0.39$) about 100 meters lower within the SCL. The enhanced correlations within this relatively narrow altitude range show that vertical velocity exerts its primary influence on surface humidity through processes operating near the top of the subcloud layer.

Figure 4 shows (a) entrainment rate E (light blue) and vertical velocity W from 100 m below the top of the SCL (dark blue), both in mm s$^{-1}$; (b) surface $\delta$ $\delta$D in ‰; and (c) water vapor mixing ratio in g kg$^{-1}$. Each point represents a 6-hour average centered on a HALO circle time. The time series shows coherent variability across all of these parameters. Periods of enhanced entrainment (positive E values) consistently coincide with more negative $\delta$D and lower mixing ratio, as seen most





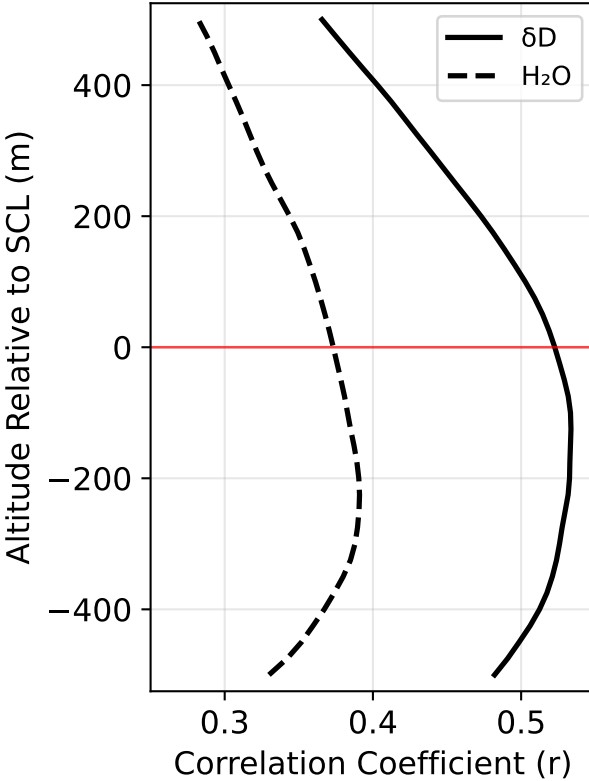

**Figure 3.** Vertical velocity correlation profiles with isotopic composition and humidity near subcloud layer height. Correlation coefficients between vertical velocity (W) and water vapor $\delta$ D (solid black line) and water vapor mixing ratio (dashed black line) are shown as functions of altitude relative to the top of the subcloud layer (SCL, red horizontal line at 0 m). The analysis spans $\pm$500 m around the top of the SCL, which is defined as the height where virtual potential temperature first exceeds its density-weighted mean from 100 m by 0.2 K. Both variables show peak correlations within the subcloud layer, with $\delta$D exhibiting stronger coupling to vertical motion than humidity.

clearly around January 26. Conversely, periods of weak entrainment or stronger mesoscale ascent (negative E, positive W)
correspond to less negative $\delta$D and higher mixing ratios, evident around January 24 and February 1. Entrainment brings dry, isotopically depleted air from above the SCL into the surface layer, reducing humidity and driving $\delta$D toward more negative values. Periods of reduced entrainment or mesoscale upward motion allow the boundary layer to maintain higher humidity and preserve the enriched isotopic signatures characteristic of ocean evaporation.

This is further illustrated in Figure 5, which shows the relationship between vertical velocity W (x-axis) and entrainment
rate E (y-axis), with points colored by surface $\delta$D values and sized according to water vapor mixing ratio. The scatter plot shows a systematic organization of boundary layer states across the E-W parameter space. Points in the upper left quadrant (negative W, high E) represent conditions of strong entrainment combined with downdrafts, characterized by more negative $\delta$D values and low mixing ratios. Points toward the lower right (positive W, low E) indicate periods of weak entrainment and





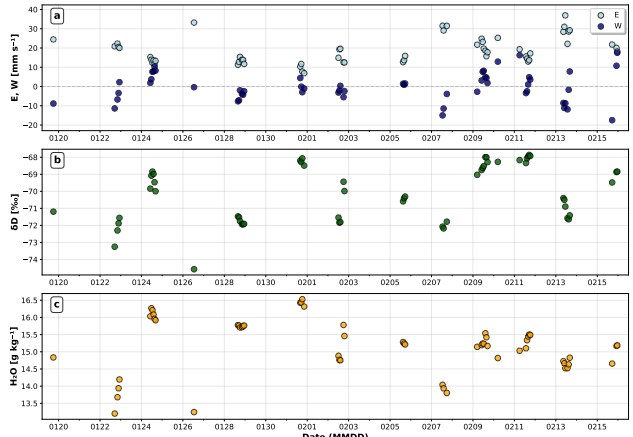

**Figure 4.** Time series of atmospheric variables during the EUREC4A campaign. (a) Entrainment rate E (light blue) and mesoscale vertical velocity W from 100 m below the top of the SCL (dark blue) in mm s$^{-1}$, showing the competing effects of horizontal dry air mixing and convective vertical motion. The dashed gray line indicates zero. (b) Surface boundary layer $\delta$D in ‰, measured by ship-based Picarro analyzer. (c) Water vapor mixing ratio H$_2$O in g kg$^{-1}$ from matched HALO-Picarro observations. All data points represent 6-hour averaged values spatially matched within HALO flight circles. The time axis shows dates in MMDD format during January-February 2020. Data demonstrate the temporal evolution of entrainment-convection competition and its effects on boundary layer moisture and isotope signatures.

stronger mesoscale ascent, which are associated with higher $\delta$D values and higher mixing ratios. This figure demonstrates
that E and W are not entirely independent but instead define a continuum of boundary layer mixing states. The color gradient from dark (depleted) to light (enriched) $\delta$D values follows a clear diagonal pattern from high E, negative W conditions toward low E, positive W conditions. Similarly, marker sizes increase along this same trajectory, indicating that periods of reduced entrainment and upward motion coincide with both isotopic enrichment and enhanced humidity.

A more quantitative understanding of these relations is illustrated in Figs. 6 and 7. Figure 6 shows the joint dependence
of isotopic composition and humidity on both entrainment and vertical velocity. Panel (A) shows $\delta$D values across the E-W parameter space, with contour lines representing predictions from a joint linear regression model that includes both E and W as predictors. The isotopic field exhibits a systematic gradient from depleted values ($\delta$D $\approx$ -73‰) at high entrainment rates and negative vertical velocities to enriched values ($\delta$D $\approx$ -68‰) at low entrainment and positive vertical velocities. Panel (B) shows the corresponding humidity field, where water vapor mixing ratios decrease from approximately 16.5 g kg$^{-1}$ at low E
and high W to 13.5 g kg$^{-1}$ at high E and low W. The orientation of the contours in both panels shows that $\delta$D is more sensitive to W than the total mixing ratio, which is more sensitive to entrainment.

Figure 7 further quantifies the counteraction effect of W on E through residual analysis. This approach first removes the linear effect of entrainment alone, then quantifies how vertical velocity correlates with the remaining variance. The $\delta$D residuals (Panel A) show a strong positive correlation with W ($r = 0.464$), indicating that upward motion systematically counteracts
entrainment-driven depletion. The counteraction efficiency is defined as the ratio of the W regression slope to the E regression




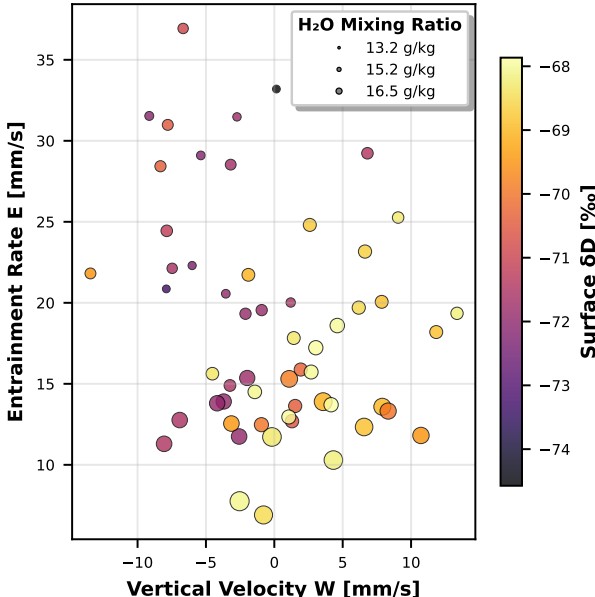

**Figure 5.** Relationships between vertical velocity (W), entrainment rate (E), water vapor mixing ratio, and water vapor $\delta$D during EUREC4A. Scatter plot shows W extracted at optimal altitude (567.7 m) versus entrainment rate E, with circle size proportional to water vapor mixing ratio ($H_2O$) and color representing surface $\delta$D. Data points represent individual dropsonde circles matched with ship-based isotope measurements.

slope and reaches 1.19 for $\delta$D, meaning that 1 mm s$^{-1}$ of upward motion counteracts the isotopic effect of 1.19 mm s$^{-1}$ of entrainment. For water vapor (Panel B), the counteraction is weaker, with $r = 0.270$ and an efficiency of 0.30. Mesoscale vertical velocities exerts a much stronger control on isotopic composition than on humidity itself. The efficiency greater than unity for $\delta$D indicates that mesoscale processes dominate over entrainment in determining surface isotopic signatures, while the

much lower efficiency for humidity reflects the more equitable relationship between vertical motion and entrainment in setting the water vapor concentrations in the boundary layer. While these correlations exceed the noise threshold identified earlier, we note that the measurement uncertainty represents a large fraction of the observed $\delta$D variance. Thus, the precise values of the regression slopes and efficiencies should be interpreted with caution, even though the qualitative pattern of a stronger isotopic than humidity response is robust.

These results demonstrate that mesoscale circulations significantly modulate the entrainment-driven changes in boundary layer humidity and composition, consistent with the findings of Vogel et al. (2022) and George et al. (2023). The isotopic measurements provide quantitative constraints on the relative importance of these competing processes, with mesoscale vertical motions capable of fully offsetting or even reversing entrainment effects on boundary layer moisture characteristics.



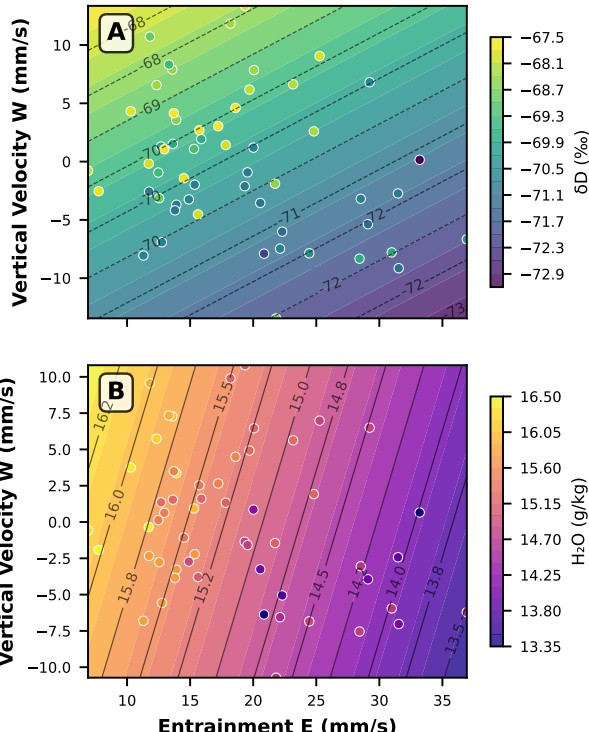

**Figure 6.** Joint entrainment-vertical velocity dependencies in isotopic composition and humidity fields. (A) Contour plot showing water vapor $\delta$D as a function of entrainment rate E and vertical velocity W, with observations overlaid as colored points. Contour lines represent predictions from the joint E–W regression model. (B) Water vapor mixing ratio (H2O) dependencies on the same E–W parameter space, using a different color scheme to distinguish variables.

## 3.1 Mixed Layer Model

To better understand the processes responsible for the asymmetric responses of $\delta D$ and $q$ to entrainment and mesoscale vertical motion, we implemented a steady-state flux-form mixed-layer model (Appendix A). The model represents the boundary layer as a single, well-mixed slab in which water vapor and its isotopic composition are determined by the balance between surface evaporation, turbulent entrainment from the cloudy layer, and large-scale vertical motions. Surface evaporation is parameterized using Craig–Gordon theory, and entrainment is prescribed as an independent flux of mass and isotopes from the overlying air.

Mesoscale vertical velocity influences the model both directly, by modulating the net mass budget of the mixed layer, and indirectly, by altering the effective entrainment rate. This simple framework captures the observed contrast in sensitivity, with $\delta D$ responding much more strongly than $q$ to variations in mesoscale vertical motion, thereby providing a mechanistic interpretation of the asymmetries seen in the observations.





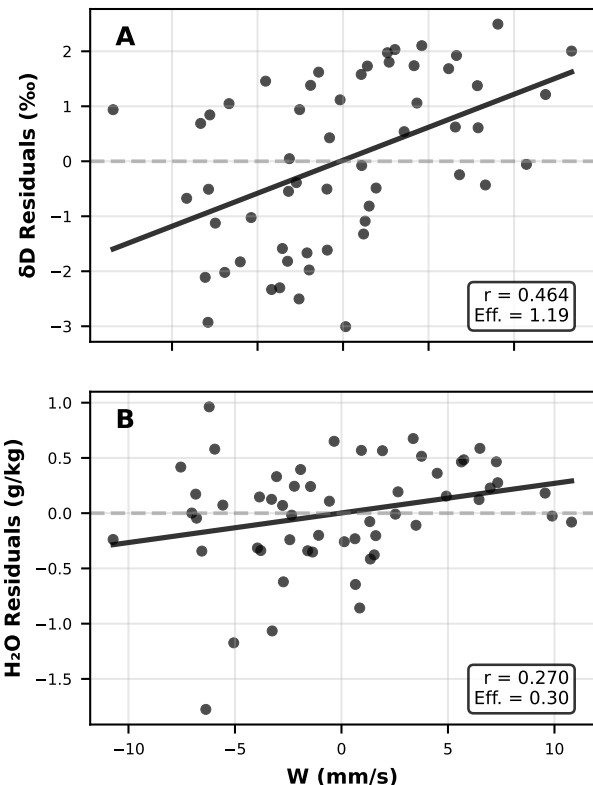

**Figure 7.** Residual analysis of vertical velocity counteraction effects on isotopic composition and humidity. (A) $\Delta D$ residuals after removing entrainment-only effects plotted against vertical velocity W at optimal altitude. Points show individual HALO circle observations, with the regression line indicating the pure counteraction effect of W. The correlation coefficient (r) and counteraction efficiency (Eff.) quantify how effectively vertical motion counteracts entrainment-driven isotopic depletion. (B) Water vapor mixing ratio (H2O) residuals after removing entrainment-only effects plotted against vertical velocity W.

Figure 8 shows the sensitivity of modeled boundary-layer humidity ($q$) and isotopic composition ($\delta D$) to mesoscale vertical
velocity $W$, expressed in standardized units to facilitate comparison. For each variable, anomalies are computed relative to the model state at $W = 0$ and then normalized by the corresponding standard deviation from the observational campaign ($\sigma_q = 0.832$ g kg$^{-1}$, $\sigma_{\delta D} = 1.94$ ‰). Expressing the results in these $\sigma$ units allows a direct comparison of the relative strength of the $W$-induced changes in $q$ and $\delta D$ despite their different physical units and variances. A slope of unity in these units would correspond to a change equal to one observed standard deviation per 1 cm s$^{-1}$ change in $W$.
The curves in Figure 8 exhibit a clear asymmetry in the modeled responses. For $W > 0$ (mesoscale ascent), both $q$ and $\delta D$ increase, while for $W < 0$ (descent) they decrease. However, the magnitude of the response in $\delta D$ is much larger: the model yields a linear sensitivity of about 3.72 $\sigma_{\delta D}$ per cm s$^{-1}$, compared to only 0.49 $\sigma_q$ per cm s$^{-1}$ for $q$, a ratio of roughly 7.5. This difference reflects the fact that $\delta D$ is sensitive not only to changes in total water mass but also to shifts in the isotopic



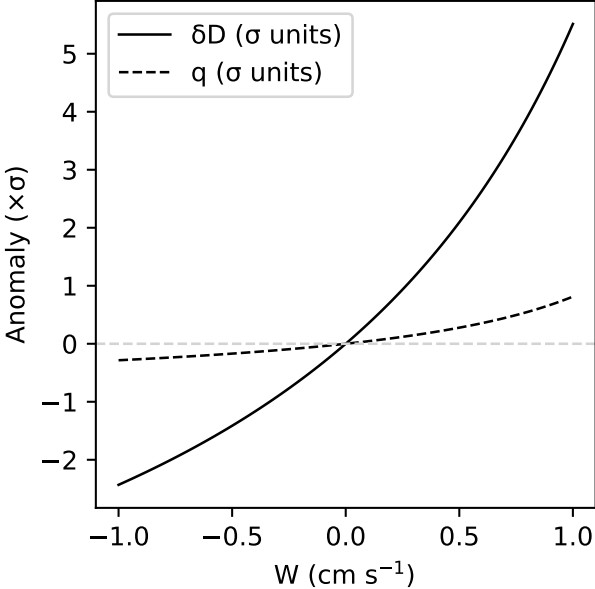

**Figure 8.** Sensitivity of boundary-layer mixing ratio (q) and isotopic composition ($\delta D$) to mesoscale vertical velocity from the flux-form mixed-layer model. Curves show anomalies of $\delta D$ (solid) and $q$ (dashed) expressed in observational $\sigma$ units versus vertical velocity $W$ (cm,s$^{-1}$), at fixed entrainment $E = 20$,mm,s$^{-1}$.

composition of the entrained air. In the model, mesoscale ascent reduces the relative contribution of isotopically depleted CL air, driving $\delta D$ upward, while mesoscale descent enhances the entrainment influence and lowers $\delta D$. Because $q$ is controlled solely by the net mass fluxes, its fractional changes are smaller.

This modeled behavior parallels the observational results: in both, the isotopic composition responds much more strongly to mesoscale vertical motion than does humidity, and the sign of the response is consistent with the balance between surface evaporation and entrainment from the CL. The $\sigma$-unit scaling underscores that the stronger isotope response is not an artifact of measurement units, but reflects a genuinely greater fraction of the observed variability being explained by $W$ in $\delta D$ than in $q$. This asymmetry emerges naturally from the model physics, without requiring any ad hoc tuning, and provides a simple mechanistic explanation for the patterns seen in the data.

## 4 Discussion

Our analysis shows that mesoscale vertical motions in the trades strongly modulate marine boundary layer moisture by counteracting entrainment-driven drying. Water vapor isotopologues reveal this process with much greater sensitivity than humidity alone, highlighting an asymmetric response in which water vapor $\delta D$ responds much more strongly to changes in mesoscale vertical velocities than the total mixing ratio. A flux-form mixed-layer model reproduces this behavior, demonstrating that $\delta D$





responds about 7.5 times more strongly than humidity to mesoscale vertical motion. These findings underscore the critical role
of mesoscale circulations in shaping humidity and isotopic composition in the trade-wind regime. Recall that throughout this
study, $E$ refers only to entrainment from the cloudy layer (CL) into the subcloud layer (SCL) and it does not include exchange
between the CL and the free troposphere (FT).

Our findings can be placed in the context of the analytical and LES-based framework of Risi et al. (2020). In their simula-
tions, the primary mechanism depleting near-surface vapor was the export of isotopically enriched air by convective updrafts,
with downdrafts and rain evaporation providing additional contributions. They argued that the classical amount effect emerges
because stronger large-scale ascent steepens vertical isotopic gradients, thereby enhancing the efficiency of updrafts and down-
drafts in depleting the subcloud layer. In contrast, our EUREC$^4$A analysis highlights the sensitivity of $\delta D$ to mesoscale vertical
motions within the trade inversion layer: shallow mesoscale updrafts counteract entrainment-driven import of depleted cloudy-
layer air and thus enrich the subcloud layer, while mesoscale downdrafts enhance the isotopic depletion. Both perspectives
underscore the role of vertical motions in setting near-surface isotopic composition, but at different scales and regimes: large-
scale ascent and deep convection in Risi et al. (2020), versus shallow mesoscale overturning circulations within the trades in
this study. Together, these results suggest that the isotopic composition of boundary layer vapor is highly sensitive to the scale
and vertical structure of vertical motions, which can act either to enrich or to deplete depending on the dynamical context.

Our results provide mechanistic closure for the shallow mesoscale overturning circulations (SMOCs) identified by George
et al. (2023) and the counteraction mechanism proposed by Vogel et al. (2022). Vogel et al. (2022) refuted the mixing-
desiccation hypothesis of Sherwood et al. (2014) by showing that mesoscale vertical motions and entrainment contribute
comparably to subcloud-layer budgets, but with opposite signs for their impact on humidity. They concluded that stronger
mixing does not desiccate cloud base because mesoscale circulations counteract the drying expected from entrainment. Mean-
while, George et al. (2023) provided the first direct observational evidence of SMOCs in the trade-wind layer, showing robust
dipoles in divergence between the subcloud and cloud layers, spatial scales of $\sim$100–200 km, and ubiquitous coverage ($\sim 58\%$
of a $10° \times 10°$ domain). They demonstrated that SMOCs amplify moisture variance at cloud base, with strong anti-correlation
between subcloud divergence anomalies and $q$ at cloud base (Pearson $r \approx -0.67$), and proposed that convergence branches
reduce entrainment drying efficiency in ascending regions, driving bottom-heavy moisture variance.

By exploiting water vapor isotopologues, we bridge these complementary findings and provide a missing process-level attri-
bution. Our $\delta D$ measurements serve as direct tracers of source contributions, showing that mesoscale ascent maintains enriched
isotopic signatures characteristic of oceanic evaporation, while entrainment introduces isotopically depleted air from the CL.
This renders the qualitative compensation mechanisms of both studies directly observable in boundary-layer composition.
Quantitatively, we show that mesoscale ascent offsets entrainment-driven depletion of $\delta D$ with an efficiency of 1.19, meaning
that 1 mm s$^{-1}$ of upward motion more than cancels the isotopic effect of 1 mm s$^{-1}$ of entrainment. For humidity, by contrast,
the counteraction efficiency is only 0.30, thereby making the isotopic imprint a far more sensitive tracer of SMOC dynamics
than humidity alone.

Our analysis extends both studies by localizing the coupling mechanisms in the vertical. We show that the strongest correla-
tions between vertical velocity and both $\delta D$ and humidity occur just below the subcloud layer top, pinpointing where mesoscale





circulations leave their imprint on near-surface composition. This vertical localization was not resolved in either previous study
and sharpens the physical understanding of how SMOCs modulate the boundary layer. The isotopic approach also reveals that
rising branches of SMOCs carry an ocean-evaporation signature into the subcloud and cloud-base layers, while descending
branches are associated with entrained air from the CL, yielding enhanced $\delta D$ contrast between branches that far exceeds the
corresponding $q$ contrast.

   When scaled by their observed standard deviations, $\delta D$ responds about 7.5 times more strongly to mesoscale vertical mo-
tion than does humidity, highlighting a novel asymmetry in the system's response to mesoscale variability. Small increases
in entrainment lead to relatively large isotopic depletions with only modest drying. By mapping $\delta D$ and $q$ jointly onto the
entrainment–vertical velocity (E–W) space, we demonstrate a diagonal organization very similar to the SMOC moisture
anomaly structure described by George et al. (2023), but with stronger isotopic amplitude, confirming and quantifying the
proposed moisture-variance mechanism.

   Our flux-form mixed-layer model provides mechanistic closure that bridges the conceptual frameworks of both studies. By
parameterizing an effective entrainment frequency $\varepsilon_{\text{eff}} \propto (E - \gamma W)$, the model captures the observed asymmetry between
$\delta D$ and $q$ without ad hoc tuning and reproduces the bottom-heavy variance patterns identified by George et al. (2023). This
framework demonstrates that the greater sensitivity of isotopes arises naturally from the stronger contrast between $\delta$ values of
surface and CL sources relative to their humidity differences.

   Thus, while our results broadly confirm the conclusions that mixing–desiccation fails in the trade-wind regime and that
SMOCs fundamentally modulate boundary layer moisture, they advance our understanding by: (1) quantifying counteraction
efficiencies, (2) providing source-specific attribution of moisture variability through isotopic tracers, (3) localizing the cou-
pling mechanisms in the vertical, and (4) offering a simple yet physically grounded model framework. Together, these findings
sharpen the refutation of the mixing–desiccation hypothesis, deepen our mechanistic understanding of SMOC-moisture cou-
pling, and provide a pathway for incorporating isotope constraints into parameterizations of shallow convection and cloud
feedbacks.

   Our findings provide extensions to the theoretical frameworks for MBL water vapor isotopic composition established by
Benetti et al. (2018) and more recent empirical work by Galewsky et al. (2022) and Risi et al. (2019). While Benetti et al.
(2018) demonstrated that their MBL-mix model successfully reproduces isotopic observations by incorporating both evap-
orative flux variability and mixing with isotopically depleted free-tropospheric air, our study shows that mesoscale vertical
velocities introduce an additional layer of complexity that fundamentally modulates these mixing processes. The counteraction
efficiency we observe, where 1 mm s$^{-1}$ of upward motion counteracts the isotopic effect of 1.19 mm s$^{-1}$ of entrainment for
$\delta$D, suggests that mesoscale circulations can completely overwhelm the traditional two-endmember mixing framework during
periods of strong vertical motion. This finding extends beyond the MBL-mix model's scope by demonstrating that the propor-
tion of water vapor from the LFT (the $r$ parameter in Benetti et al. (2018)) is not simply determined by entrainment rates but is
dynamically modulated by mesoscale vertical velocities on timescales much shorter than typically considered.

   The asymmetric response we document between $\delta$D and humidity to mesoscale forcing provides quantitative support for the
processes outlined in Galewsky et al. (2022), while revealing mechanisms that operate across different scales and environments.



Our observations from EUREC4A show $\delta$D responding 7.5 times more strongly than humidity to mesoscale vertical motion (in $\sigma$-units), which parallels Galewsky et al. (2022)'s finding of stronger isotopic than humidity responses to decoupling in stratocumulus-topped boundary layers. However, our EUREC4A results demonstrate that this asymmetry extends to the trade cumulus regime and operates on mesoscale rather than boundary layer decoupling timescales. The vertical localization we observe, with strongest correlations occurring just below the subcloud layer top, provides the missing mechanistic link between the surface isotopic signatures and the shallow mesoscale overturning circulations (SMOCs) identified by George et al. (2023). This suggests that the local sources of water vapor invoked by Galewsky et al. (2022) must be understood within the context of mesoscale circulation patterns that can transport locally evaporated, but subsequently fractionated, water vapor between different levels of the boundary layer.

Our results further advance the understanding developed by Risi et al. (2019) regarding the relative importance of surface fluxes versus atmospheric mixing in controlling MBL isotopic composition. While Risi et al. (2019) demonstrated that boundary layer mixing processes could explain isotopic variability without invoking large-scale horizontal transport, our study reveals that the mixing itself is not a passive process but is actively modulated by mesoscale dynamics. The efficiency with which vertical velocity counteracts entrainment effects (efficiency > 1.0 for $\delta$D) indicates that during periods of mesoscale ascent, the traditional closure assumptions fundamentally break down. The MBL cannot be treated as a simple two-endmember system during active mesoscale periods, as the vertical motion effectively changes the isotopic composition of the entrained endmember by redistributing water vapor that has undergone different degrees of processing within the boundary layer. This finding reconciles the apparent contradiction between Risi et al. (2019)'s emphasis on local processes and the need to explain isotopic depletions that exceed what simple surface-atmosphere exchange can produce.

The integration of our results with these previous studies reveals that the marine boundary layer isotopic composition emerges from a balance between surface evaporation (Craig–Gordon effects), entrainment mixing (Benetti et al. (2018) framework), boundary layer decoupling processes (Galewsky et al. (2022) mechanisms), and mesoscale circulation modulation (our new contribution). Rather than invalidating previous work, our study demonstrates that these assumptions hold only during quiescent periods and must be replaced by a more dynamic framework during periods of significant mesoscale activity. That mesoscale processes dominate over entrainment in determining surface isotopic signatures while showing weaker control over humidity suggests that isotopic measurements provide a uniquely sensitive probe of mesoscale dynamics that cannot be detected through humidity observations alone.

## 5 Conclusions

The goal of this study was to quantify how mesoscale vertical motions modulate marine boundary layer moisture and isotopic composition in the trade wind regime, and to test the mixing–desiccation hypothesis using water vapor isotopologue observations from EUREC[4]A.

We found:



1. Water vapor $\delta$D responds approximately 7.5 times more strongly to mesoscale vertical velocity variations than humidity itself (when normalized by observed standard deviations), with $\delta$D showing a linear sensitivity of 3.72 $\sigma_{\delta D}$ per cm s$^{-1}$ compared to 0.49 $\sigma_q$ per cm s$^{-1}$ for specific humidity.

2. Mesoscale upward motion counteracts entrainment-driven isotopic depletion with an efficiency of 1.19, meaning 1 mm s$^{-1}$ of vertical velocity more than cancels the isotopic effect of 1 mm s$^{-1}$ of entrainment, while the counteraction
305        efficiency for humidity is only 0.30.

3. The strongest correlations between vertical velocity and both $\delta$D ($r \approx 0.52$) and mixing ratio ($r \approx 0.39$) occur within $\pm 200$ m of the subcloud layer top, pinpointing where mesoscale circulations exert their primary influence on surface composition.

4. Periods of enhanced entrainment consistently coincide with more negative $\delta$D values and lower mixing ratios, while
310        periods of mesoscale ascent correspond to less negative $\delta$D and higher humidity, demonstrating systematic organization of boundary layer states across the entrainment–vertical velocity parameter space.

5. A steady-state flux-form mixed-layer model reproduces the observed asymmetric responses, confirming that isotopic composition is more sensitive to changes in source mixing ratios than bulk humidity, thereby providing mechanistic closure for the counteraction effects observed in the trade wind regime.

While the relatively small natural variability in $\delta$D during the campaign means that the precise quantitative sensitivities should be interpreted with caution, the convergence of multiple diagnostics and agreement with the mechanistic model give confidence in the qualitative conclusion: isotopic composition provides a far more sensitive tracer of mesoscale circulations than humidity alone.

*Data availability.* EUREC4A water vapor isotope data from R/V Meteor and HALO dropsonde data from the JOANNE dataset are publicly
available through the EUREC4A data portal at https://eurec4a.eu/data.

## Appendix A: Flux-Form Mixed-Layer Model Formulation

### A1  Model Overview

We use a steady-state, flux-form mixed-layer model to represent the moisture and isotopic budgets of the subcloud layer (SCL) in the tropical marine boundary layer. The model is spatially homogeneous in the horizontal, has a fixed SCL height $h$, and
resolves only bulk mean quantities: the specific humidity $q_{BL}$ and the isotopic composition $\delta_{BL}$ of boundary-layer water vapor. The SCL exchanges water and isotopes with three reservoirs:

1. **Surface source:** evaporation from the ocean, with specific humidity $q_{flux}$ and isotopic composition $\delta_{flux}$.





2. **Cloudy-Layer (CL) source:** entrainment of air from above the top of the SCL, with $q_{\text{FT}}$ and $\delta_{\text{FT}}$.

3. **Ventilation sink:** large-scale advection or lateral mixing at rate $\lambda$ (s$^{-1}$).

The model tracks the competition between these fluxes as modulated by the entrainment rate $E$ and the mesoscale vertical velocity $W$. We introduce an "effective" entrainment frequency $\varepsilon_{\text{eff}}$ to capture the modulation of entrainment by mesoscale motions.

## A2  Governing Equations

Let $q_{\text{BL}}$ be the boundary-layer specific humidity in g kg$^{-1}$, and $R_{\text{BL}}$ the isotopic ratio (D/H) in absolute units. The steady-state
moisture budget is

$$0 = \lambda\left(q_{\text{flux}} - q_{\text{BL}}\right) + \varepsilon_{\text{eff}}\left(q_{\text{FT}} - q_{\text{BL}}\right), \tag{A1}$$

where

$$\varepsilon_{\text{eff}} = \max\left(\frac{E - \gamma W}{h}, 0\right). \tag{A2}$$

Here $E$ and $W$ are in m s$^{-1}$, $h$ in m, and $\gamma$ is a dimensionless "counteraction efficiency" relating vertical velocity to effective
entrainment reduction. In the calculations presented here, $\gamma = 1.0$. Negative values of $\varepsilon_{\text{eff}}$ are clipped to zero.

Solving (A1) for $q_{\text{BL}}$:

$$q_{\text{BL}} = \frac{q_{\text{flux}}\lambda + q_{\text{FT}}\varepsilon_{\text{eff}}}{\lambda + \varepsilon_{\text{eff}}}. \tag{A3}$$

For the runs shown, $q_{\text{flux}}$ is approximated by the saturation mixing ratio at the sea-surface temperature and pressure, $q_s(\text{SST}, p)$, computed with the Tetens formula for $e_s(T)$ and

$$q_s = \frac{\varepsilon e_s}{p - e_s}, \quad \varepsilon = 0.622. \tag{A4}$$

## A3  Isotopic Balance

Let $R = 1 + \delta/1000$ be the conversion from $\delta$ notation (in ‰) to absolute isotopic ratio normalized to VSMOW. Denote $R_{\text{flux}}$ and $R_{\text{FT}}$ as the source isotopic ratios corresponding to $\delta_{\text{flux}}$ and $\delta_{\text{FT}}$.

The steady-state isotopic mass balance is

$$0 = \lambda\left(q_{\text{flux}}R_{\text{flux}} - q_{\text{BL}}R_{\text{BL}}\right) + \varepsilon_{\text{eff}}\left(q_{\text{FT}}R_{\text{FT}} - q_{\text{BL}}R_{\text{BL}}\right). \tag{A5}$$

Solving for $R_{\text{BL}}$:

$$R_{\text{BL}} = \frac{\lambda q_{\text{flux}}R_{\text{flux}} + \varepsilon_{\text{eff}}\, q_{\text{FT}}R_{\text{FT}}}{(\lambda + \varepsilon_{\text{eff}})\, q_{\text{BL}}}. \tag{A6}$$



The model output $\delta_{\mathrm{BL}}$ is then

$$\delta_{\mathrm{BL}} = 1000\,(R_{\mathrm{BL}} - 1). \tag{A7}$$

For the surface source isotopic composition $\delta_{\mathrm{flux}}$, the model by default uses the Craig–Gordon formulation, which depends on SST, relative humidity, and pressure. With the campaign-state values (SST = 27.3°C, RH = 0.716, $p = 101325$ Pa), this yields $\delta_{\mathrm{flux}} \approx -74.7‰$.

## A4    Parameter Values Used

For the runs shown in the main text figures:

– $h = 650$ m

   – $\gamma = 1.0$

   – $q_{\mathrm{FT}} = 3$ g kg$^{-1}$

   – $\delta_{\mathrm{FT}} = -150\,‰$

   – $\delta_{\mathrm{flux}}$ from Craig–Gordon with SST = 27.3°C, RH = 0.716, $p = 101325$ Pa, giving $\approx -74.7\,‰$

– $\lambda = 0.10$ day$^{-1}$ (converted to s$^{-1}$ in computations)

   – $E$ and $W$ explored in ranges: $E \in [10, 35]$ mm s$^{-1}$, $W \in [-10, 10]$ mm s$^{-1}$

*Author contributions.*  JG conceived the study, analyzed the data, developed the mixed-layer model, and wrote the manuscript. SL processed the isotopic data and contributed to the analysis. Both authors contributed to the interpretation and editing of the manuscript.

*Competing interests.*  The authors declare no competing interests.

*Acknowledgements.*  The authors used artificial intelligence (ChatGPT, OpenAI, 2025) to assist with coding the mixed-layer model and with drafting some initial text. All code, results, and manuscript content were thoroughly checked, validated, and revised by the authors, who take full responsibility for the accuracy and interpretation of the work. We acknowledge the EUREC4A campaign organizers and participants, particularly the crew of R/V Meteor and the HALO aircraft operations team. This work was supported by the U.S. National Science Foundation under grant AGS-1853353. We thank the German Research Foundation (DFG) for support of the EUREC4A campaign
under grant 264907654.



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
