# Peer review of "Mesoscale modulation of marine boundary layer water vapor isotopologues during EUREC4A"

_EGUsphere, 2025_

## Referee Comment (RC1)

**ACP - 2025 - Galewsky and Los**

Galewsky and Los present an analyses of stable water vapor isotope measurements taken during the EUREC4A campaign to demonstrate the role that the mesoscale has to play in modulating moisture heterogeneities near the surface. They show that an isotopic signature is more prominent while diagnosing the mesoscale effect compared to humidity values alone. They use observational analyses and a simple-physics steady state model to demonstrate how the mesoscale might be modulating these moisture and isotopic differences. The study is very timely and address a prominent question that two recent studies (Vogel et al., 2022 and George et al., 2023) have left in their wake while highlighting the importance of the mesoscale. This study also taps into the immense potential of using isotopes for understanding moisture pathways in the atmosphere - something traditional measurements are limited in. If their results can be substantiated (with suggested corrections here to their own analyses) and with more evidence from future investigations, this study lays down important groundwokr in showing how the mesoscale mediates moisture - an open question in the shallow cumulus' research community, with important implication for cloud feedback and climate sensitivity studies.

That said, I have a few major as well as minor issues with some aspects of the study, which I have listed here below (and due to a lack of time, have been unfortunately unable to digest down into more concise text).

**MAJOR COMMENTS:**

**1. Apparent AI Hallucination:**

- a. The Galewsky et al (2022) study cited multiple times throughout the paper has an incorrect full refernce at L395. I believe the correct citation should be Galewsky, J., Jensen, M. P., & Delp, J. (2022). Marine boundary layer decoupling and the stable isotopic composition of water vapor. Journal of Geophysical Research: Atmospheres, 127, e2021/ID035470. <a href="https://doi.org/10.1029/2021/ID035470">https://doi.org/10.1029/2021/ID035470</a>. However the study cited, is a fictional one. I could find no study of the given title or with the given combination of authors. The DOI leads to a paleoclimate glaciation study (that the DOI exists probably is why any automated checks from the journal might have missed this). This seems like an unfortunate consequence of using Al-powered tools that are known to hallucinate. However, the authors are responsible for the final product and such mistakes shouldn't be carried over in to it, not least when the full citation in the study includes both the present authors. I haven't made similar checks for all studies listed in the references, but I would request that the authors do so.
- 2. Data discrepancy and unclear specifications:
  - a. L110: "The measurement uncertainty for 6-hour averaged isotope data (1.24% for δD) represents approximately 65% of the observed natural variability, constraining our ability to detect weak atmospheric signals. Consequently, our analysis focuses on the strongest and most robust correlations that exceed this measurement noise threshold" What exactly does this mean for the data considered / statistics shown in this study? Were only those circles considered whose correspoding δD values lay outside the 1.24% range from the campaign mean?
  - b. Figure-3: For these correlations, how were the ship-derived properties sampled? Were these averaged over the time-period that HALO took to fly the circle? Circle-time was ideally obtained through the flight-segmentation files provided via Konow et al (2021) to be consistent with the W measurements.
  - c. Figure-4: (a) It is unclear how the circles have been chosen for the E, W. Most days show fewer than 6 circles, which was the case for a typical HALO flight. However, 0209 shows 7 data-points whereas there were only 6 circles flown for that flight. There was no HALO flight on 0210, but there is a data point in the figure. Such discrepancies are very surprising.
  - d. L121: I don't understand the choice of a 6-hour average centered on circle time. One circle takes one hour. A typical HALO flight had 3 consecutive circles, followed by a roughly 1.5 non-circle excursion, followed again by 3 consecutive circles. So, in a period of 7-8 hours, there are 6 circle measurements available. Having 4-6 data points per day in this plot means that many data points likely have an overlap of 4-5 hours. So, the 6-hourly average doesn't make sense, the autocorrelation between the data-points would be very high.
- 3. Statistical analyses and interpretations thereout:
  - a. Figure-5 and L129-L138: I can see that the E-W space indeed shows a range of BL moisture states and I understand the physical reasoning put forward by the authors. However, I am unconvinced that there is a "clear diagonal pattern" for δD from high E, negative W to low E, positive W. With a similar level of agreement, one could also argue that ignoring the 5 points of E > 30 mm/s, there exists a diagonal pattern from low E, negative W to higher E, positive W with more depleted points and more enriched points in the lower-left and upper-right, respectively. This would go against the relationship that the authors describe in the text. If the authors could please provide some quantification (e.g. a regression coefficient) for these relationships, then it would be more convincing.
  - b. Figure-6: In continuation with my understanding that the correlations are poor in Figure-5, I think that the multi-linear regression (MLR) model also doesn't show convincing results in its agreement with the observed data points. Panel-(B) seems to be doing slightly better, but the large spread in both panels indicates to me that the MLR is not a good predictor for either of the variables. Please also provide the coefficients and the residual errors that come out of the MLR.

**4. Counteraction efficiency:**

- a. L147-L164 and Figure-7: I really like the idea and intent behind this analysis, i.e. studying the competing effects of E and W on δD, and is a timely effort indeed. However, the analysis doesn't really separate the effects of E and W. The residuals are computed by first removing the linear regression coefficient of E to explain D. Next these residuals are regressed against W and an r-value of 0.464 is obtained. This residual regression is only applicable if we consider that E and W are uncorrelated. That's not the case. Vogel et al (2022)'s Fig-4g shows that E and W have a negative correlation (r=-0.25 at 1h and -0.35 at 3h, and could be stronger if we go with the 6-hour choice in the study). A better way of getting the "true" independent linear relationships that E and W have with δD would be getting their coefficients from the multi-linear regression already done for Figure-6. Here the coefficients for E would assume effect with constant W and that for W would assume constant E.
- b. I'd also like to add a note about the interpretation of the "counteraction efficiency", which the authors define as the ratio of the slope of regression of W to that of E. In the present analysis, the slope of regression for W is with the residuals whereas for E, it is with δD. At least this is how I understood L150 which follows the description regression with residuals. So, the slopes are not really comparable (note also my previous comment about other inconsistencies in performing such an analysis). However, even if both coefficients are from a regression against δD (say from a multi-linear regression), there should be some normalization done before computing the coefficients so as to account for the mean and range of the predictors E and W (as shown in L366). Otherwise, 1 mm/s of E cannot be comparable to a 1 mm/s of W and the physical conclusions for counteraction interpreted here will not really be that observed in nature.

**5. Precipitation events:**

- a. Precipitation and cold-pools were both important mesoscale-modifying events during the EUREC4A campaign as shown by Touzé-Peiffer et al., 2022 and by Radtke et al, 2022 as well as Radtke et al, 2023. Certain flights (evident from Touze-Peiffer's study) during the campaign did sample both precip and cold-pools. These data-points could be ones that will show a contrasting signal in the moisture-W feedback, because with cold-pools you would have higher moisture but low values of W.
- b. The cold-pool's effect also might be responsible for the vertical localization being around 500 m, because as such in the absence of cold-pool cases, W seems to scale linearly upwards from the surface to around SCL top. This would mean that the correlation shouldn't necessarily change from near-surface to SCL top. However, with cold-pools the values of h will go below 400 m, thus preferentially improving correlation scores above the 400 m which is also what seems to be the case in Figure-3 in the present study. If the correlation is checked by controlling for data-points associated with precipitation and/or cold-pools (as suggested above), then the study can solidify why the vertical localization happens at that altitude.
- c. The effect of precipitation and cold-pools is missing from the entire discussion, which have a big role to play in setting the BL moisture as well as modulating the properties of the mesoscale. See for example studies like Vogel et al (2021) and Alinaghi et al (2025). Isotope measurements are significantly impacted by the presence of precipitation and resulting evaporation-induced cold pools. Not including a discussion of how such events have impacted these findings would be remiss.

**MINOR COMMENTS:**

- Throughout the text, consistency between EUREC4A and EUREC4A
- L53: "atmospheric properties" instead of "circulations"

- L60: HALO measurements and flight strategies for EUREC4A are described in detail in Konow et al (2021), and hence should ideally be cited here for the reader who
  might be interested in more details.
- L63: instead of "mesoscale circulation", say "area-averaged kinematic". The latter is more apt and precise.
- L66: and that stationary in the time-period that is taken to sample that space.
- L70: 70 EUREC4A-circles over how many days? The number of days would indicate roughly the different synoptic scale meteorological conditions that were present for
  the sampling of the mesoscale circulations
- L76 (or L63): At one of the earliest mentions of "divergence" in the text, it should be explicitly stated that this is "horizontal divergence of wind velocity (hereafter referred to simply as divergence)". Otherwise, it remains unclear the divergence of which quantity is being talked about and if the divergence is 3D or horizontal. Same with vorticity, but this term doesn't appear later.
- L78: "h" is later (e.g. Section-3) referred to as SCL height keep terminology consistent.
- L90: Seeing as these terms (E & h) hold quite some weight in later discussions, it would help the reader greatly to know what were the mean (+- std dev) values through the campaign. It provides good context for interpreting values discussed later.
- · L93: "sampling rate" and not "resolution"
- L108: Please specify where the RH and SST measurements are from. I'm assuming the RV Meteor has in-house instruments, but please specify if this is the case. Also what height and depth were these measured, respectively?
- L118: "vertical velocity exerts its primary influence on surface humidity through processes operating near the top of the subcloud layer." Not enough data yet shown to imply causation.
- L119: How should one interpret the fact that  $\delta D$  shows consistently greater correlation with W compared to q?
- L124 and L125: E values are always positive. The distinction in the text is unclear when the authors refer to the enhanced and weak entrainment periods. Maybe the authors intend to show this as anomaly of E from campaign-mean? If so, that would make sense with the peripheral text in the paragraph.
- L125: No HALO flight on February 1, the text should state January 31 instead.
- L128: Whereas the arguments in this paragraph make sense and could be indicative of what the data shows, there is an inherent assumption here that the surface
  fluxes are consistent for all the days. This might not be a valid assumption because the estimation of E here considers it directly proportional to the surface buoyancy
  fluxes (SBF). This means that enhanced E also means enhanced surface fluxes (likely increased wind) given unchanging stability between the sub-cloud and cloud
  layers. Vogel et al (2022)'s Fig-4b indeed shows that E is very well correlated with SBF and less so with the buoyancy jump. Thus, I believe that the anti-correlation
  between E and δD cannot be simply explained without controlling for the surface flux conditions, which also play in a role in how fast or slow they can enrich nearsurface water vapor.
- Moreover, it needs to be stated that the effect that W has on q or δD of the BL is not a direct physical effect unlike that explained for the effect of E. In assumptions here
  of a well-mixed boundary layer, whereas both W and E change h, only E can change the BL properties (such as q and δD). W cannot change q and δD because of the
  well-mixed assumption. Any relationship that is observed between W and the BL properties should be thought of as a consequence of a self-reinforcing feedback.
- Well-mixed assumption. Any relationship that is observed between W and the BL properties should be thought of as a consequence of a self-reinforcing feedback.
   Figure-5 caption: Why is 567.7 m the optimal altitude for W? Is that where correlation peaks in Figure-3? If so, please specify either in caption or somewhere in text.
- Figure-6: The contour line labels in panel (A) don't make any sense with repeating numbers. My first guess that the plot might accidentally contain rounded values for the contour line labels also failed seeing the irregular spacing between those integer repetitions, e.g. one -69, three -70s, one -71 and three -72s... The contour line labels in panel (B) look fine.
- L165 and other mentions of "asymmetry" here on: May I suggest "varying" as a more appropriate term than asymmetric? The relationships are both symmetric wrt E and W, that's what the linear analyses done before in fact assumes, i.e. a symmetric response. The difference is in the strength (slope) of the relationships of q and δD, and hence the suggestion "varying relationship" instead of "asymmetrical relationship". The authors themselves clarify this in L180-182.
- · Mixed-layer model:
  - I like the "counteraction efficiency" term that the authors have introduced to try and quantify the competing effects of W and E. The use of it, however, in the mixed-layer model is not the same as that interpreted from the observations (see my comment regarding L147-164 and Figure-7). Also, why not use its value obtained from the regressions instead of v = 1.0?
  - The steady-state aspect of the model simplifies calculations indeed, however the authors should verify the validity of such an assumption when comparing to the observations. See Figure-4 where one can see both q and isotopic composition vary quite a lot within just 8-10 hours if the data points correspond to circle times from HALO. Thus, at scales being discussed here, the 0 in Eq A.1 is not valid. The authors should therefore note (preferably around L171) that the MLM helps in crudely placing relative importances of the processes, but there can be major deviations at the observed scales in nature due to the transient state of the boundary layer (i.e. the large spread in Figures-5 and 6).
- L180 192: This is indeed an interesting demonstration of possible convective-precipitation processes that the self-reinforcing nature of W modulates. That it comes out of simple physics is a joy to see.:) It shows the potential of how physical hypotheses can be derived from stable isotope measurements.
- L194: "strongly" is based on the 1.19 counteraction efficiency, I believe. So, this might change.
- L202 L212: I enjoyed this discussion and the different findings in deep v/s shallow convective regimes, although I did not understand the detailed argument of Risi et al (2020). Regardless, I would request the authors to specify the spatial scales that Risi et al (2020) consider. In the last sentence of the paragraph, I am unsure what is meant by the "vertical structure of vertical motion" setting the contrasting observations. Do the authors mean that the difference in dynamics of shallow convection (keeping BL moist) and deep convection (bringing cold and dry air aloft to the surface) results in the contrasts?
- L244: First sentence of paragraph is essentially a repetition from earlier.
- The discussion misses findings of LES studies such as Bretherton and Blossey, Janssens et al (2022) and Janssens et al (2024) which indicate more of a top-heavy moisture variance and almost no moisture differences due to the mesoscale in the sub-cloud layer. The findings from the present study (if held true) is important observational evidence that contradicts the LES findings.
- L259 L260... I suggest the following change for more clairification. "... our study shows that mesoscale vertical velocities, through self-reinforcing mechanisms, introduce an additional layer of complexity that fundamentally modulates ..."
- For the discussion and conclusion sections, here I don't point out separately parts that discuss the counteraction efficiency (especially its quantification) or other aspects that might need to change based on the authors' response to my previous major comments.

**References:**

- a. Alinaghi, P., Jansson, F., Blázquez, D. A., and Glassmeier, F.: Cold pools mediate mesoscale adjustments of trade-cumulus fields to changes in cloud droplet number concentration, Atmos. Chem. Phys., 25, 6121–6139, https://doi.org/10.5194/acp-25-6121-2025, 2025.
- b. Bretherton, C. S., & Blossey, P. N. (2017). Understanding mesoscale aggregation of shallow cumulus convection using large-eddy simulation. Journal of Advances in Modeling Earth Systems, 9, 2798–2821. https://doi.org/10.1002/2017MS000981
- c. Janssens, M., De Arellano, J. V. G., Van Heerwaarden, C. C., De Roode, S. R., Siebesma, A. P., & Glassmeier, F. (2023). Nonprecipitating shallow cumulus convection is intrinsically unstable to length scale growth. Journal of the Atmospheric Sciences, 80(3), 849-870.
- d. Janssens, M., George, G., Schulz, H., Couvreux, F., & Bouniol, D. (2024). Shallow convective heating in weak temperature gradient balance explains mesoscale vertical motions in the trades. Journal of Geophysical Research: Atmospheres, 129(18), e2024JD041417.
- e. Konow, H., Ewald, F., George, G., Jacob, M., Klingebiel, M., Kölling, T., Luebke, A. E., Mieslinger, T., Pörtge, V., Radtke, J., Schäfer, M., Schulz, H., Vogel, R., Wirth, M., Bony, S., Crewell, S., Ehrlich, A., Forster, L., Giez, A., Gödde, F., Groß, S., Gutleben, M., Hagen, M., Hirsch, L., Jansen, F., Lang, T., Mayer, B., Mech, M., Prange, M., Schnitt, S., Vial, J., Walbröl, A., Wendisch, M., Wolf, K., Zinner, T., Zöger, M., Ament, F., and Stevens, B.: EUREC4A's HALO, Earth Syst. Sci. Data, 13, 5545–5563, <a href="https://doi.org/10.5194/essd-13-5545-2021">https://doi.org/10.5194/essd-13-5545-2021</a>, 2021.
- f. Radtke, J., Naumann, A.K., Hagen, M. & Ament, F.(2022) The relationship between precipitation and its spatial pattern in the trades observed during EUREC4A. Quarterly Journal of the Royal Meteorological Society, 148(745), 1913–1928. Available from: <a href="https://doi.org/10.1002/qi.4284">https://doi.org/10.1002/qi.4284</a>
- g. Radtke, J., Vogel, R., Ament, F., & Naumann, A. K. (2023). Spatial organisation affects the pathway to precipitation in simulated trade-wind convection. Geophysical Research Letters, 50, e2023GL103579. https://doi.org/10.1029/2023GL103579

---

## Referee Comment (RC2)

**Review of Galewsky and Los 2025**

November 11, 2025

This article documents how humidity and water vapor isotopic composition of the subcloud layer in the tradewind region vary with entrainment and mesoscale vertical motion. It uses measurements from the EUREC4A campaign and a simple analytical model of the subcloud layer.

The motivation for this study is to contribute to the debate on the validity of the mixing-desiccation mechanism for the sub-cloud layer in the trade-wind regions (Sherwood et al 2014), especially regarding the role of mesoscale circulations (Vogel et al 2022). This is an important topic for the scientific community interested in cumulus cloud feedbacks.

The article argues that the isotopic composition is more sensitive to mesoscale vertical motion than to entrainement, whereas the humidity is more sensitive to entrainment than to mesoscale vertical motion. I found that the argument for this based on EUREC4A observations was convincing. However, I was not convinced by the physical mechanisms to explain this behavior. I think the analytical model fails to capture this behavior, and this is my major comment.

Except from this comment, my comments are only minor. The paper is well written and illustrated.

**1 Major comment: the analytical model fails to capture the observed behavior at the core of the article**

The observed behavior at the core of this paper is that the isotopic composition is more sensitive to mesoscale vertical motion than to entrainement, whereas the humidity is more sensitive to entrainment than to mesoscale vertical motion. Using the notations of the article, it means that:

$$d\delta_{BL}/dW > d\delta_{BL}/dE$$

$$dq_{BL}/dW < dq_{BL}/dE$$

Fig 6 is very convincing to prove this.

Fig 7 also confirms this behavior (if I understand well, "Eff." is a non-dimensional parameter that reflects  $(d\delta_{BL}/dW)/(d\delta_{BL}/dE)$  or  $(dq_{BL}/dW)/(dq_{BL}/dE)$ ?)

Any successful analytical model should capture this behavior. However, this is not the case for the analytical model proposed here. Rather, an analysis of the equations yields  $d\delta_{BL}/dW$ )/ $(d\delta_{BL}/dE) = (dq_{BL}/dW)/(dq_{BL}/dE)$ .

The article argues that  $\delta D_{BL}$  responds 7.5 times more to W than  $q_{BL}$ . However, this is an artifact of normalizing by the observed standard deviation. There is no good reason to do this normalization, because the standard deviation may have nothing to do with the impacts of W and E that we are investigating. In addition, I don't think that just showing that  $(d\delta_{BL}/dW)/\sigma_{\delta BL} > dq_{BL}/dW)/\sigma_{qBL}$  is useful for the whole argument of the paper, which is to investigate the interplay between E and W. So we really need to compare the response to E and to W.

Therefore, I recommend not artificially normalizing by the observed standard deviation. Non-dimensional numbers can be obtained by calculating rations such as  $(d\delta_{BL}/dW)/(d\delta_{BL}/dE)$  or  $(dq_{BL}/dW)/(dq_{BL}/dE)$ , which roughly correspond to the slopes of the diagonal contours on Fig 6.

Showing the E-W plot for the modeled  $q_{BL}$  and  $\delta_{BL}$  as in Fig 6 would be sufficient to show the failure of the model.

The failure of the model is expected from its equations:  $q_{BL}$  and  $\delta_{BL}$  depend on  $\epsilon_{eff}$  which depends on E-W. Therefore, this model does not allow to separate the impacts of E and W.

I had some fun playing around with equations to try to come up with equations that separate E and W and in which  $d\delta_{BL}/dW)/(d\delta_{BL}/dE)$  would be different from  $(dq_{BL}/dW)/(dq_{BL}/dE)$ . Actually, I couldn't. I even

tried a two-layer model, it failed as well. Maybe I didn't try hard enough, but I expect that making  $q_{BL}$  and  $\delta D_{BL}$  respond differently to E and W is not so obvious.

Maybe some alternative hypotheses should be considered to explain that  $\delta_{BL}$  responds more strongly to W than to E whereas  $q_{BL}$  responds more strongly to E than to W. One hypothesis: mesoscale ascent favors the development of cumulus clouds, whose rain drops evaporate in the subcloud layer? It has a strong enriching effect, but only a moderate moistening effect. A simple analytical model that includes this effect would capture the observed behavior.

**2 Minor and detailed comments**

- I don't agree with the sentences arguing the model captures the asymmetric behavior, consistent with my major comment: 19, most of the discussion and conclusion
- 1 109: "similarly constrained": what does constrain mean here?
- l 111: "constraining" -> limiting?
- 1 121: remove one  $\delta$
- Fig 3: could it be useful to also plot h as a function of time? I was wondering to what extend variations in h alone could contribute to the observed correlation pattern. Would it be useful to plot some W profiles as well? To see where the W extrema are relative to h?
- Fig 4: make larger labels
- 1 136: "clear diagonal pattern": I cannot see it clearly on Fig 5. Rather, I can see a q gradient with E, i.e. horizontal pattern. This impression is confirmed by the nearly vertical lines on Fig 6b.
- l 155: "more equitable": I disagree, Fig 5 and 6 rather show that E dominates for W, whereas it is more "equitable" (i.e. diagonal pattern) for  $\delta$ .
- 1 163: "moisture characteristics" -> "moisture  $\delta D$ " only?
- Fig 7:  $\Delta$ ->  $\delta$  in the caption
- l 176: I recommend not doing this normalization, consistent with my major comment. Same Fig 8. Replace by something more comparable to Fig 6.
- l 201: FT: I think earlier it was written that the entrained air comes from the CL. Make notations consistent? Same l 264 and throughout appendix A.
- 1 209: "large-scale ascent" -> "convective updrafts": in Risi et al 2020, the export of enriched air is through convective scale updrafts.
- 1 222: "bottom-heavy variance": what does it mean? Same 1 246
- 1 250-252: I disagree with (4) consistent with major comment. I'm also not convinced by (1) and (2). For (1): can you come up with a number? For (2): can you be more specific?
- Appendix A3: if  $R_{flux}$  is given by Craig and Gordon, then it should depend on  $R_{BL}$ . It's important to account for this feedback because it attenuates  $\delta$  variations in the boundary layer, so that you need a more strongly enriching process when W > 0 in your model than if you ignore this feedback. When deriving the equation, you will end up with a Merlivat-like equation for  $R_{BL}$ , as in Risi et al 2020.

---

## Author Comment (AC1)

**Author Response to Referees**

**Manuscript:** *Mesoscale modulation of marine boundary layer water vapor isotopologues during EUREC4A*
**Authors:** Galewsky & Los

The reviewer comments substantially improved the clarity, rigor, and physical interpretation of the manuscript, and we appreciate their feedback. Below we reproduce the referee comments (italicized) and respond point-by-point. Line numbers refer to the revised manuscript unless otherwise noted.
* * *
**Response to Referee #1**

**1. Apparent AI hallucination / citation error**

*"…The Galewsky et al. (2022) study cited multiple times has an incorrect full reference… This appears to be a hallucination…"*

**Response:**
The referee is correct. The reference was incorrect and has been fully corrected. The correct citation is now:

Galewsky, J., Jensen, M. P., & Delp, J. (2022), *Marine boundary layer decoupling and the stable isotopic composition of water vapor*, JGR-Atmospheres, 127, e2021JD035470.

All references were re-checked manually to ensure no further errors remain.
* * *
**2. Measurement uncertainty and data inclusion (L110)**

*"…What exactly does this mean for the statistics shown? Were only circles with $\delta D$ values outside the 1.24‰ range considered?"*

**Response:**
No filtering was applied based on $\delta D$ uncertainty. The value 1.24‰ represents the **1$\sigma$ absolute accuracy of a 2-hour mean $\delta D$ measurement** from the shipboard Picarro system, expressed in per mil (‰), not percent. All HALO and P-3 circle-matched observations are included in the analysis.

The purpose of stating this uncertainty is to quantify the **noise floor** relative to the observed campaign variability ($\sigma \delta D \approx 1.94$‰). Measurement uncertainty therefore contributes scatter but does not alter which points are included. We now clarify this

explicitly in the text and emphasize that we focus interpretation on gradients and contrasts that exceed this noise level.
* * *
**3. Figure 3 timing and sampling consistency**

*"…Were ship-derived properties averaged over the HALO circle time? Ideally circle-time should be obtained through Konow et al. (2021) segmentation…"*

**Response:**
All timing is keyed to the **JOANNE Level-4 circle definitions**, which use the Konow et al. (2021) flight-segmentation framework. Shipboard $\delta D$ and $H_2O$ values are averaged within ±2 hours of each JOANNE circle time and spatially matched to the circle footprint. Vertical velocity profiles come from the same JOANNE dataset and the same circle identifiers. We have clarified this explicitly in the Methods.
* * *
**4. Figure 4 discrepancies (number of circles, Feb 1)**

*"…0209 shows 7 points… 0210 shows a point despite no HALO flight…"*

**Response:**
These points correspond to **P-3 circles**, not HALO. The original text incorrectly implied HALO-only usage. We now explicitly state that both **HALO and P-3** dropsonde circles are used throughout and label this clearly in the text and figure captions.
* * *
**5. Averaging methodology (6-hour vs ±2-hour)**

*"…The choice of a 6-hour average is confusing…"*

**Response:**
The referee is correct. This was legacy wording from an earlier analysis. All averaging is ±2 hours around each circle time. The incorrect "6-hour" language has been removed everywhere.
* * *
**6. Diagonal structure in Figure 5**

*"…The 'clear diagonal pattern' is unconvincing without quantification…"*

**Response:**

We agree and removed the word "clear." We now quantify the relationship explicitly. Using all 56 circle means, the fitted relationship is:

$$E = -0.25(\pm 0.24)\,W + 18.6, \quad p = 0.044,$$

demonstrating a modest but statistically significant negative tilt. This quantification is now reported.
* * *
**7. Multilinear regression performance (Figure 6)**

*"…Please provide coefficients and residual errors…"*

**Response:**

Done. We now report full regression coefficients, p-values, MAE, RMSE, and $R^2$ for both $\delta D$ and $H_2O$. The spread in the figures is consistent with the reported $R^2$ values and is discussed explicitly.

- $\delta D \sim E + W$: coefficients [const –68.74, E –0.071, W +0.100]; $R^2$ = 0.357. Coefficient p-values: E p≈0.013, W p≈0.00027 (both significant at 5%). Residual MAE = 1.15‰; RMSE = 1.33‰.

  - $H_2O \sim E + W$: coefficients [const 16.72, E –0.083, W +0.019]; $R^2$ = 0.633. Coefficient p-values: E p≈$2.5 \times 10^{-11}$, W p≈0.040. Residual MAE = 0.365 g/kg; RMSE = 0.472 g/kg.
* * *
**8. Counteraction efficiency and residual analysis**

*"…Residual analysis assumes E and W are uncorrelated…"*

**Response:**

We agree this needed clarification. We now define counteraction efficiency **exclusively from standardized coefficients of the joint E–W regression**, which yields the **partial effects accounting for E–W covariance (r ≈ –0.27)**. The residual plot is retained only as a visualization of the partial relationship and is explicitly described as such.
* * *
**9. Normalization and unit bias**

*"…1 mm s$^{-1}$ of E is not comparable to 1 mm s$^{-1}$ of W…"*

**Response:**
Agreed. We revised the definition of counteraction efficiency to use **standardized regression coefficients** ($|\beta W/\beta E|$), which is unitless and scale-free. Physical cancellation in mm s$^{-1}$ is now discussed separately using unstandardized slopes, and the distinction is made explicit throughout.
* * *
**10. Precipitation and cold pools**

*"...The effect of precipitation and cold pools is missing from the discussion..."*

**Response:**
We added a dedicated paragraph addressing cold pools and precipitation, citing Touzé-Peiffer et al. (2022) and Radtke et al. (2022, 2023). We note that the strongest isotopic–W correlations occur above the shallowest cold-pool-affected layers, suggesting that the signal primarily reflects deeper mesoscale overturning rather than near-surface cold-pool dynamics.
* * *
**11. Minor comments and line edits**

All suggested minor edits, terminology corrections, figure clarifications, and reference additions (Konow et al., LES studies, etc.) have been implemented. We thank the referee for the detailed attention.
* * *
**Response to Referee #2**

**Major comment: analytical model failure**

*"...The analytical model fails to capture the core observed behavior..."*

**Response:**
This critique was pivotal and correct. The original model formulation could not reproduce the observed separation between $\delta D$ and q sensitivities. We therefore **revised the model** to include **hydrometeor–vapor isotopic exchange**, motivated directly by the referee's suggestion. With this addition, the model reproduces the observed E–W contour structure and the stronger isotopic sensitivity. The model description and Appendix were rewritten accordingly.
* * *
**Normalization critique**

*"…The 7.5× sensitivity is an artifact of normalization…"*

**Response:**
We agree and removed the problematic normalized sensitivity plot. The model is now evaluated by comparing **E–W contour structures directly**, consistent with Figure 6.
* * *
**Steady-state assumption**

*"…The steady-state assumption is crude…"*

**Response:**
We now explicitly state that the mixed-layer model is **diagnostic**, not predictive, and intended to clarify relative sensitivities rather than capture transient evolution. This limitation is now acknowledged in both the Methods and Discussion.
* * *
**Minor comments**

All minor text edits, figure label changes, terminology corrections, and reference additions have been incorporated. We also clarified the definition of "bottom-heavy variance" following George et al. (2023).
* * *
**Closing statement**

We again thank both referees for their careful and constructive reviews. The manuscript is substantially improved as a result, particularly in the interpretation of counteraction efficiency, the analytical model formulation, and the discussion of mesoscale–isotope coupling.